# The Effect of Playing Gateball Sports on Older Chinese People’s Wellbeing in the Context of Active Aging—Based Mediation of Social Capital

**DOI:** 10.3390/ijerph191912254

**Published:** 2022-09-27

**Authors:** Xinze Li, Ningxiao Tang, Ronghui Yu, Hongyu Jiang, Hongwei Xie

**Affiliations:** 1School of Physical Education, Hunan University of Science and Technology, Xiangtan 411199, China; 2Physical Education School, Jimei University, Xiamen 361000, China

**Keywords:** active ageing, sports participation, social capital, happiness, gateball

## Abstract

Participation in gateball sports may improve the well-being of older individuals in the context of active aging. However, the mechanisms of the effect need a social viewpoint. A random sample of 337 valid data points was gathered from seven cities in the Chinese province of Hunan. Structural equation modeling, fuzzy set qualitative comparative analysis, and other techniques examined the standard structure and causal links between involvement in gateball sports, social capital, and elderly people’s well-being. According to structural equation modeling, playing gateball sports may improve elderly people’s well-being, and social capital may influence this link to some extent. The engagement in gateball sports and social capital work in concert to promote well-being, according to a qualitative comparative study of the fuzzy sets that identified four patterns of “A, B, C, and D” antecedent constructs that do so in elderly people.

## 1. Introduction

Population aging is a big issue for governments all over the globe. For example, according to 2020 estimates from the seventh national census, the number of senior citizens in China has reached 264 million, accounting for 18.7% of the total population, a 5.44% rise over the preceding decade. China is predicted to enter an era of deep aging this year and will become the world’s most severely aged nation in the future, directly addressing the pressures of aging. Participation in sports significantly contributes to healthy aging and is a public health development priority. With increased epidemiological research and the implementation of new health promotion models, the importance of sports engagement in encouraging elderly people is receiving international attention. Sports engagement enhances elderly people’s immune systems and increases psychological well-being. According to some academics, sports engagement helps decrease negative psychological states such as stress, worry, and isolation induced by emerging public crises. In the new global pandemic, appropriate physical involvement is advocated as a vital strategy for controlling the psychological well-being of elderly people. According to Professor Morris, one of the pioneers of epidemiology, finding methods to encourage physical engagement is arguably the “best bargain” available [1].

With age, people’s physical functioning and control decline, and they are more vulnerable to negative symptoms such as anxiety, loneliness, sleeplessness, and social situations that may set off a chain reaction of negative emotions. The worldwide emergence of the new coronary pandemic posed a significant danger to human life and hastened the aging process. Although home isolation may successfully limit the spread of the pandemic, it has the potential to harm elderly people’s mental health. Some studies have shown that during an pandemic, home isolation may have a detrimental influence on people’s mental health, with elderly people having much poorer levels of mental health than younger ones [2,3,4]. According to several studies, the senior population is at significant risk of social isolation during the pandemic, which may lead to substantial effects such as physical obesity, poor health, and mental disintegration [5,6,7]. In China, 83.3% of the elderly need care, 11.7% are unsatisfied with their living circumstances, and 33.1% are depressed. They are especially prone to irreparable mental health repercussions when confronted with widowhood, empty nests, unemployment, disease, and incapacity amid the new global pandemic. As a result, enhancing the mental health of the elderly is a critical livelihood initiative in China, and happiness is a crucial indicator of mental health.

A national plan to actively deal with population aging should be adopted, according to the “Proposal of the Central Committee of the Communist Party of China on Formulating the 14th Five-Year Plan for National Economic and Social Development and the Visionary Goals for 2035”. Among these, “how to improve the well-being of the elderly” is an essential aspect of active aging. The well-being of elderly people is derived chiefly from emotions and contentment, i.e., their subjective perception of the quality of life. Therefore, sports involvement, as an effective strategy to improve elderly people’s well-being, offers happy feelings, and promotes life satisfaction. According to the third national mass sports survey findings, gateball ranks fourth in participation among sports activities in which elderly people aged 65 and up participate, trailing only walking, running, and qigong, and is one of the better-run sports among elderly people. Gateball is a sport where a little ball is whacked with a mallet over a small goal. It is a team sport and a leisure activity popular among senior Chinese people because of its low exertion, easy rules, and social enjoyment. Some researchers believe that gateball activities can be used as an adjunct medical tool and that long-term participation in them can help suppress the rise in cholesterol levels in elderly people’s bodies, effectively preventing diseases such as hypertension and hyperlipidemia and contributing to physiological satisfaction [8,9,10]. In addition, some researchers think that gateball may assist the elderly in stabilizing their emotions, efficiently eliminating loneliness in life, and boosting psychological fulfillment [11,12,13]. It has been established that gateball exercises not only build the body but also assist the elderly in reducing stress, extending their social network, and improving their well-being. Based on this, it is critical to investigate the influence of sports engagement on the well-being of elderly people via the lens of gateball activities.

Sports engagement has been found to improve the well-being of elderly people, although the link is complicated by several circumstances. Several mediating factors might explain the influence of sports engagement on the well-being of elderly people. Some studies have adopted a dynamic approach, claiming that engagement in sports enhances elderly people’s well-being by decreasing negative emotions of loneliness, while participation in sports may promote leisure pleasure, which contributes to elderly people’s subjective well-being [14,15,16]. Some research shows that sports involvement may improve mental well-being by improving psychological resilience and that it can also improve psychological capital and, consequently, life satisfaction in older individuals [17,18,19]. However, few studies have looked at the relationship between gateball sports participation and elderly people’s well-being, as well as the mechanisms that influence it from a social (e.g., social capital) standpoint. While social capital has been a popular notion in China in recent years, this research sought to investigate the complicated causal link between gateball sports engagement and elderly people’s well-being from the standpoint of social capital. Although regression analysis and structural equation modeling are the most commonly used methods for examining causal connections between variables, they have limitations in evaluating the combined impacts of numerous variable components on well-being. This research used a fuzzy set qualitative comparative analysis methodology to understand better the causal linkages between variables, which mixes symmetric and asymmetric methodologies to improve the interpretation of the data.

To summarize, the research framework of “gateball sports participation, social capital, and happiness” was constructed using gateball activities as the entry point, integrating structural equation modeling and fuzzy set qualitative comparative analysis, and revealing the impact of gateball sports participation and social capital through empirical analysis of the effects of gateball sports participation on happiness and the role of social capital. A combination of the structural equation model and the fuzzy set qualitative comparative analysis was utilized to investigate the influence of goalball sports involvement on well-being and the function of social capital. The combination of the two methods not only confirms the linear correlation but also explains the complexities of the causal relationship, which aids in clarifying the pathway of the impact of gateball sports participation on well-being, which is essential for improving the well-being of the elderly [20,21].

## 2. Literature Review and Research Hypothesis

Sports participation is a socially engaging activity and an organized and regulated athletic behavior that aims to increase the general public’s quality of life and physical and mental health via physical exercise and recreational consumption. Kenyon makes a point about the presence of both direct and indirect sports involvement in his understanding of the role of sport [22], which is summarized and summarized by Lu Yuanzhen [23]. Direct sports participation refers to the physical engagement of the masses in sport, whilst indirect sports participation refers to the participation of the people in sport via forms such as consumers. Therefore, individual engagement in sports is a process aimed at boosting physical and mental health via physical activity and recreational consumption.

### 2.1. Sports Participation and Well-Being of Elderly People

Happiness arises from action and is obtained organically via direct or indirect involvement in the activity [24]. According to this painstaking research, sports involvement may help others interact and collaborate more effectively and produce a feeling of enjoyment in the act, favoring the continuous development of pleasant emotions [25]. According to specific research, involvement in sports enhances physical and mental health, boosts life satisfaction, and consistently improves the quality of life [26]. As a result, participation in scientific and sensible sports becomes a crucial initiative utilized by elderly people to enhance their feeling of well-being.

The effects of physical activity on the well-being of elderly people have been the subject of research on the impact of physical engagement on the well-being of elderly people. As per studies, elderly people who participate in regular physical exercise are significantly happier than those who do not [27]. Zuo discovered that modest home physical exercise for older individuals during the current pandemic was linked to better physical and mental health and a stronger feeling of well-being [28]. In terms of indirect sports engagement, elderly individuals may watch sports to relieve the loneliness associated with empty nests and improve their well-being. Alternatively, elderly people might meet their behavioral desires for emotional pleasure and mental calm by participating in sports and volunteering [29,30,31]. Furthermore, hosting significant athletic events such as the Olympics and championships may boost elderly people’s feelings of belonging and well-being in the short term [32]. Notably, based on the UTAUT approach, Hsu elucidates the mechanism of action that significantly contributes to the subjective well-being of elderly people in Taiwan Province via long-term gateball involvement.

Based on the above, we proposed Research Hypothesis 1.

**Hypothesis** **1.**
*Participation in gateball sports greatly influences elderly people’s well-being.*


### 2.2. Identification of the Concept of Social Capital and How It Is Measured

In contrast to human and economic capital, which joined the current sociological body of knowledge in the late 1980s and sparked a change in the notion of analysis from group to connectivity, social capital is a broad term. Even though they all begin with the framework of connectivity, social capital is a very broad notion, and the development of this idea by various researchers differs significantly, primarily containing the three main viewpoints and views listed below.

Coleman is the most prominent proponent of the social structure theory. Coleman [33] contended that social capital is “created as a social relationship or a social structure of relationships”, that it is “some aspect” of the social structure, and that it is “a collection of resources inherent in family relationships and community social organization that are useful for a child or young person’s cognitive or social development”. It is a set of resources inherent in family interactions and community social organizations that benefit a child’s or young person’s cognitive or social development.

Putnam is the most prominent proponent of social organization theory. Putnam [34] defined social capital as “social organizational qualities, such as networks, norms, and social trust, that encourage coordination and cooperation for mutual benefit”, and “reciprocal behavior arises when trust is generalized and institutionalized”. Individual social capital, or social capital in the context of social resources, emphasizes the various flows of resources that exist in individual social relationships and their effects; collective social capital, or social capital in the context of social structures and social organizations, emphasizes the generation of social capital and its effects on all actors in a broader context such as neighborhoods, communities, regions, and nations.

Potts and Linnan are the most prominent representatives of social capital theory. “Social capital is the capacity of people in a network or larger social organization to employ limited resources”, writes Potts [35]. “Social capital is the resources contained in social networks that actors acquire and employ in their activities”, writes Linnan [36]. In essence, social capital is a transferable resource between social actors and is contained in a network of ties between social actors.

The notion of social capital has taken a long time to create and popularize, and it still lacks agreement [37]. Despite the fact that most studies split social capital into individual and communal dimensions, there is still significant disagreement between them. On the one hand, social capital is a contextual quality that displays social cohesiveness via processes such as mutual trust, social norms, and mutual gain. On the other hand, social capital is defined as a social network that may supply resources such as social support, knowledge, and reputation. The former is concerned with the collective’s “contextual” impact on the individual, while the latter is concerned with measuring structures at the individual and collective levels. As a result, social capital is further separated into cognitive social capital and structural social capital.

The number and quality of connections between elderly people, i.e., the amount to which elderly people are acquainted with each other on a participatory basis, might indicate structural social capital in the context of active aging. This has objectivity and emphasizes the degree of social engagement. Social involvement is the process of acquiring information and exchanging it with others. It represents the quality of elderly people’s relationships. As a result, social engagement and involvement are the primary markers for gauging elderly people’s structural social capital.

The subjective social trust metric may assess cognitive and social capital. Social trust is a kind of social capital ingrained in interpersonal relationships and supports volitional behavior—for example, resource or knowledge exchange, combining, or sharing. Cognitive and social capital provide older individuals a shared vision, attracting more like-minded people and developing a more incredible feeling of trust. As a result, in this research, social trust was selected as the primary predictor of cognitive social capital among elderly people.

### 2.3. The Role of Social Capital in Enhancing the Well-Being of Elderly People

Social capital is a unique social resource that promotes the formation of social networks and trust, hence improving the well-being of elderly people. There is a robust association between more significant amounts of social capital and well-being at the structural social capital level, with happy individuals having considerably more social contacts than sad ones. This is because structural social capital relies on social interactions to encourage elderly people to form social networks and, over time, to deepen intra-individual relationships and improve their well-being. According to Dürkheim, those who actively broaden their social networks are more socially connected and have greater levels of well-being. In addition, social trust has been demonstrated to be a significant determinant in the well-being of elderly individuals in terms of cognitive and social capacity. According to research, the more social trust, the better the requirements of the elderly are served and the longer their well-being is maintained [38]. According to Italian research, elderly people with more social capital fared better during the current global pandemic, whereas those with weaker social capital fared worse [39]. In conclusion, regardless of the study, social capital adds to the well-being of older individuals.

Based on the above discussion, research Hypothesis 2 was proposed:

**Hypothesis** **2.**
*Social capital has a considerable beneficial impact on the well-being of elderly people.*


### 2.4. The Role of Sports Participation in Enhancing the Social Capital of Elderly People

Participation in sports and social capital have a selective affinity, a dialectic between group and connectivity. Sports involvement is developed via conversation and practice, creating social capital, which expands social networks and resources through athletic grounds. This demonstrates how sports engagement may successfully boost elderly people’s social capital.

The importance of social capital for elderly people has been underlined by sociology and gerontology. On the one hand, Cumming [40] contends that loneliness in elderly people is caused by the permanent erosion of social roles, social positioning, and social networks and that sports involvement is seen as an effective means to establish social networks and reduce loneliness in elderly people. During the New Crown pandemic, studies have shown that elderly people increased their social capital by participating in virtual sports [41]. On the other hand, elderly people may successfully increase social involvement by participating in volunteer sports activities and joining sports social organizations. In other words, athletic arenas offer players a wealth of social resources that encourage social involvement. Fenton [42] contends that by connecting in the virtual world of “passport fans”, elderly people have mitigated the adverse effects of the new pneumonia pandemic and successfully improved their social capital. This shows that sports involvement is an efficient means of organizing social capital accumulation.

Seippel [43] contends that social trust is stronger among official members of sports clubs than among informal members, which is one of the prerequisites for establishing cognitive social capital. This is because complete members of gateball clubs practice, play, and live together daily, recognizing the social function of “teammates” and developing a solid relationship and an incredible feeling of trust. Other studies provide factual evidence for this since elderly people who participate in gateball obtain a sense of identification and belonging, encouraging social confidence.

Based on the above discussion, research Hypothesis 3 was proposed: 

**Hypothesis** **3.**
*Participation in gateball sports greatly benefits elderly people’s social capital.*


The effect of sports engagement on the well-being of elderly people is not only emotional and psychological but also social. Engagement in sports by elderly people in groups is considerably more likely to improve well-being than participation in sports by older individuals. This is since physical exercise in older age groups necessitates two-way contact and promotes the growth of social networks and trust. On the other hand, physical activities for elderly people tend to be lonely, missing the processes of learning, communication, encouragement, and collaboration, which are not favorable for building social relationships. Ryff [44] presents empirical evidence that group-type sports involvement (e.g., gateball, square dancing) may improve the well-being of elderly adults by increasing social contact. According to Quan, by attending gateball games and helping in sports, elderly people may not only meet more people and obtain more learning chances but also boost social involvement and social trust, which promote life satisfaction. It may be argued that involvement in gateball sports can benefit the well-being of elderly people via social capital. This leads to study Hypothesis 4. Figure 1 depicts a theoretical model of the influence of gateball sports engagement on the well-being of elderly people.

**Hypothesis** **4.**
*Social capital has a mediating role in the impact of gateball sports engagement on the well-being of elderly people.*


## 3. Research Methodology

In this research, “sports participation” refers only to gateball activities and excludes other sports.

### 3.1. Study Area and Data Sources

Gateball is a popular sport among the elderly, and it has been introduced as a new athletic activity for the Hunan National Fitness Challenge Day in 2021, based on local characteristics, to encourage the development of the national fitness policy. Previously, the event was conducted for a month in 14 cities (independent prefectures), but owing to the COVID-19 pandemic, it was only hosted in 7 cities (autonomous prefectures) in 2021. (The event was not held in 2020 due to the new pneumonia pandemic). The research team chose seven cities as geographical practice areas, including Changsha, Zhuzhou, Yongzhou, Zhangjiajie, Changde, Yueyang, and Loudi, and used a random sampling method to distribute questionnaires (completed voluntarily) only to elderly people who participated in the Gateball Challenge in each city. All subjects gave their informed consent for inclusion before they participated in the study. The study was conducted in accordance with the Declaration of Helsinki, and the protocol was approved by the Ethics Committee of Hunan University of Science and Technology.

A total of 412 paper questionnaires were issued, with 372 returned, removing invalid questionnaires with unclear or missing essential information, for a total of 337 valid questionnaires (81.8%), 156 cases (46.3%) for men, and 181 instances (53.7%) for women. The control variables in the regression study were gender, age, education level, employment status, and marital status (Table 1). 

### 3.2. Design of the Questionnaire

We investigated the importance of gateball sports engagement on well-being, particularly among older individuals, and the mediating influence of social capital in this relationship. A questionnaire was utilized to examine the association between athletic involvement, social capital, and elderly people’s well-being.

The questionnaire design was carried out in the context of active aging by relying on relevant research findings and merging the features of the gateball program. The questionnaire was constructed with Liang Deqing’s scale [45], using a 5-point Likert scale (1 being the lowest score and 5 being the highest). The Cronbach’s alpha was 0.874, the CR was 0.9431, and the AVE was 0.7353, suggesting that the study was reliable and valid (see Table 2).

The social capital questionnaire was created using Tan Yanmin’s [46] scale and included nine choices separated into social trust, social involvement, and social interaction. A 5-point Likert scale was utilized, with lower scores indicating lower social capital and higher scores indicating better social capital. Cronbach’s alpha values varied from 0.713 to 0.743, CR values ranged from 0.7807 to 0.8016, and AVE values ranged from 0.5432 to 0.5742, suggesting strong reliability and validity (see Table 2).

The happiness questionnaire was created using Liu Xiuxia’s [47] scale and was divided into three categories: life satisfaction, economic satisfaction, and subjective happiness, with a total of three options on a 10-point Likert scale (1 being the lowest score and 10 being the highest score), with higher scores indicating greater happiness. Cronbach’s alpha was 0.795, CR was 0.861, and AVE was 0.6738, showing high reliability and validity (see Table 2).

### 3.3. Common Method Deviations

When respondents fill out the questionnaire, there may be common method bias. As a result, Harman’s single factor test was adopted to eliminate the data’s common method bias. The primary component factor test was performed on the questionnaire questions included in the three dimensions of sports participation, social capital, and happiness using spss26.0 software (IBM, Armonk, NY, USA). The obtained eigenroot values were more significant than 1 for three factors, whose cumulative variance was explained by 61.983%. The variance explained for the first factor was 28.611% (50%). In conclusion, there is no severe concern about common method bias.

### 3.4. Model Suitability

Table 3 displays the fit of the models. CFI = 0.971 (>0.9), NFI = 0.951 (>0.9), and TLI = 0.971 (>0.9) are value-added fit tests; PCFI = 0.783 (>0.5), PNFI = 0.741 (>0.5) are parsimonious fit tests; and x2 = 171.322 (the smaller, the better), x2/DF = 1.839 (3.0), RMSEA = 0.027 (0.08), and GFI = 0.921 (>0.9) are absolute fitness. In summary, the model’s value-added fitness test indicator, parsimony fitness test indicator, and absolute fitness test indicator are all within the specified criteria, suggesting that it is fit.

## 4. Research Results and Analysis

### 4.1. Correlation Analysis

The correlations between the two variables of sports participation, social trust, social participation, social interaction, and well-being were tested by calculating Pearson correlation coefficients (Table 4). On the one hand, the r of sports participation and social capital was less than 0.7, and the correlations of both variables showed significant positive correlations (social trust: r = 0.339, *p* < 0.001; social participation: r = 0.305, *p* < 0.001; social interaction: r = 0.217, *p* < 0.001); on the other hand, the r of sports participation and happiness was less than 0.7, and the correlations of both variables showed a significant positive correlation (r = 0.261, *p* < 0.001).

### 4.2. Hypothesis Testing of Direct and Mediating Effects

#### 4.2.1. Tests of the Direct Role

Figure 2 depicts the structural equation model of sports involvement, social capital, and well-being, and Table 5 depicts the route coefficient test findings. It can be seen that “sports participation happiness” (*p* < 0.001) indicates that sports participation has a significant positive effect on elderly people’s happiness, and hypothesis 1 holds; “sports participation social trust” (*p* < 0.01), “sports participation social participation” (*p* < 0.01), and “sports participation social interaction” (*p* < 0.01), indicating that sports participation has a significant positive effect on the enhancement of elderly people’s social caliber. “Social participation happiness” (*p* < 0.01), “social trust happiness” (*p* < 0.001), and “social interaction happiness” (*p* < 0.001), suggesting that social capital has a strong positive influence on senior happiness improvement, and hypothesis 3 holds. In conclusion, hypotheses 1–3 were all validated.

#### 4.2.2. Testing the Role of Mediation

The Bootstrap mediation path test findings (Table 6) revealed that the direct (*p* < 0.001) and indirect (*p* < 0.001) effects were both within the 95% confidence range (excluding 0). The built structural equation model contains both direct and indirect effects, indicating that it was a partially mediated model. For instance, the mediation impact of the transport participation social capital well-being (*p* < 0.01) route varied from 0.027 to 0.029 within a 95% confidence interval (excluding 0). In summary, Hypothesis 4 was evaluated, and the model demonstrated a positive partial mediation effect with a mediated effect size (%) of 47.84% (indirect impact/overall effect).

## 5. Qualitative Comparative Analysis of Fuzzy Sets

### 5.1. Selection and Calibration of Variables

The fuzzy set qualitative comparative analysis (fsQCA) can capture the link between changes in data degree and set affiliation. FsQCA differs from regression analysis because it uses two core measurements as fitting parameters: consistency and coverage. Using fsQCA analysis enables a thorough examination of the complicated causal linkages between sports involvement, social capital, and well-being on the net impact of structural equation modeling. The extent to which pathways sharing a combination of circumstances have the same result is indicated by consistency; coverage reflects the extent to which the condition variable covers the outcome variable. In conclusion, including fsQCA analysis in confirming the net effect of structural equation modeling is advantageous in studying the influence of diverse combinations of factors on the well-being of elderly people.

In structural equation modeling, the impacts of the first two variables on the well-being of elderly people were verified. Sports involvement, social capital, and age were considered antecedent factors. Only the age factor (*p* < 0.05) influenced the well-being of elderly people among the control variables. As an antecedent variable, age was included in the fsQCA analysis.

The fsQCA 3.0 program was used to calibrate the antecedent variables of sports participation, social capital, and age to increase the interpretability of elderly people’s well-being (outcome variable). The data were calibrated or assigned variables (age is a categorical variable) using three-valued fuzzy set criteria (95% = completely affiliated; 51% or 49% = neither fully affiliated nor not affiliated; 5% = not affiliated at all).

### 5.2. Analysis of the Results of fsQCA

#### 5.2.1. Necessity Condition Analysis

Before performing the conditional grouping analysis, the antecedent variables required a need analysis (consistency and coverage). According to Table 7, the consistency of each of the antecedent variables was less than 0.9, indicating that none of the antecedent factors could be a necessary condition for the result variable alone. To summarize, the influence of a mixture of antecedent factors on the well-being of older individuals must be considered, as well as their adequate circumstances.

#### 5.2.2. Conditional Configuration Analysis

A streamlined approach adds a logical remnant to a grouping that already provides a result and continues to create a result; a complicated solution rejects an antecedent variable (logical remnant) that has already caused an effect and continues to produce a result. The core conditions that affect the result variables are antecedent factors that occur in both streamlined and complicated solutions; antecedent variables that exist exclusively in intermediate solutions are also referred to as marginal conditions.

Overall consistency was 0.917 (>0.8), and overall coverage was 0.893 (>0.5), indicating that the seven histories were sufficient combinations of conditions for the outcome variables and grouping histories with the same core conditions into the same pattern for a total of four patterns, A, B, C, and D (Table 8).

Model A relates to A1 and A2 groupings. A1: “age*sports participation*social participation*social interaction” (* denotes “and”, denotes “not”, the same below) with a consistency of 0.939 and coverage of 0.337, with high age and sports participation as the core conditions and social participation and social interaction as the marginal conditions; A2: “age*sports participation*social trust*social participation”, with a consistency of 0.911 and coverage of 0.392, with high age and sports participation as the core conditions.

Model B relates to B1 and B2 groupings. B1: “Sport participation*social trust*social engagement” with 0.863 consistency and 0.253 coverage and 0.939 consistency and 0.337 coverage, with social participation as the core condition and sport participation, social trust as the marginal condition; B2: “Social trust*social participation*social interaction” with 0.917 consistency and 0.375 coverage, with social participation as the core condition and social trust, social interaction as the marginal conditions.

Group C1 relates to Model C. C1: “Sports participation * social trust * social interaction”, with a consistency of 0.953 and coverage of 0.509, with sports participation as the core condition and social trust and social interaction as the marginal criteria.

Model D refers to D1 and D2 groupings. D1: “age*sports participation*social trust*social participation*social interaction” with a consistency of 0.961 and coverage of 0.412, with low age, social trust, and social interaction as core conditions and sports participation and social participation as marginal conditions; D2: “age*sports participation*social trust*social participation*social interaction” with a consistency of 0.883 and coverage of 0.454, with low age, social trust, and social interaction as core conditions and sports participation as marginal conditions.

#### 5.2.3. Robustness Tests

Stability tests were used to re-calibrate the anchor locations. The conditional grouping analysis was repeated after calibrating the “complete affiliation” from 0.95 to 0.75 and the “complete disaffiliation” from 0.05 to 0.25. The research revealed that the primary conditions and grouping pathways remained constant, suggesting that the findings were relatively stable.

## 6. Discussion

### 6.1. Analysis of the Positive Effect of Sports Participation on the Well-Being of Elderly People

The development of pandemics in recent years has resulted in a decline in sports engagement, a rise in the prevalence of mental disease, and a decrease in well-being. This is affects the elderly particularly severely. On the other hand, numerous cross-sectional, longitudinal studies have verified the benefits of sports involvement for the well-being of elderly people, such as square dancing, walking, and gateball. According to social cognitive theory, cognition is an individual’s assessment of their overall traits and confidence in their future ambitions. Some elderly adults have limited cognition about sports and lack drive during their first engagement, which may quickly lead to unhappiness. Participation in gateball sports is a gradual process that may be utilized to improve cognitive levels and increase endogenous motivation via practice at both direct and indirect levels, progressively enhancing elderly people’s well-being.

Gateball physical activity may unleash the body and mind and directly enhance physical and mental health, while how to preserve physical and mental health is a crucial issue in the quest for well-being for elderly people. There are parallels and variations in terms of demands between the two. Through mutual collaboration and conversation, elderly people may not only increase their confidence in one another and improve their feeling of belonging to the organization but also momentarily forget about the disagreeable aspects of their lives and boost their replenishment of positive energy. In other words, gateball exercises may help the elderly cope with unpleasant emotions. Gateball exercise may make elderly people happy since it is high in aspects that promote well-being, such as creating “pleasing” and “smooth” feelings and providing “self-confidence”, “self-esteem”, and “self-improvement”. Elderly people may reduce stress in their life by participating in deep gateball; the more often they exercise with gateball, the greater the exercise experience, resulting in more positive energy benefits. Furthermore, when playing gateball, the body secretes a range of chemicals or neurotransmitters, including dopamine, endorphins, cortisol, and serotonin, all of which have mood-regulating and relaxing effects. This study’s results are consistent with prior research that has demonstrated that gateball exercise improves the well-being of elderly people. It can be observed that gateball exercises may provide elderly people with emotional relief and a feeling of well-being. It could be because when elderly people learn gateball activities, they improve their technical and tactical skills, as well as their cognitive level, through offline exchanges and exchanges with each other; it could also be because they have the noble sentiment of giving a rose to others, which promotes the intermingling of elderly people’s emotions; or it could be because elderly people receive fitness guidance from social sports instructors, and through long-term social interaction and mutual humbling, they improve their technical and tactical skills and cognitive level.

In terms of indirect sports involvement, viewing gateball contests and consuming gateball sports may help the elderly achieve their psychological value demands, resulting in a pleasurable emotional experience. On the one hand, when elderly adults watch a game of gateball, their attention shifts from inattention to concentration, their attitude changes from neglect to attention, and their role shifts from spectator to participant, all of which are explicit representations of emotional transformation. This is supported by a study that shows that watching a gateball match triggers positive emotional changes in an individual’s life, leading to increased life satisfaction. By sharing gateball matches, elderly people can gain a sense of joy and solidarity from victory and experience and encouragement from defeat, which can lead to a change in social roles, for example, from being a spectator of gateball to participating in gateball volunteering, which continues to improve the quality of life. On the other hand, the consumption of gateball sports can satisfy the demands of the elderly for vibrant health to accomplish the goal of physical and mental leisure. Sports consumption behavior is psychologically derived from consumption motivation, which is separated into four levels: social, recreational, fitness, and challenge, all with the ultimate goal of meeting wants and realizing ideals. In other words, the higher the level of consumption of gateball sports among elderly people, the greater their motivation to act, their leisure engagement, the higher the quality of their experience, and the better they can satisfy their daily demands, thus boosting their feeling of well-being. It was evident that viewing and eating gateball may improve the quality of life and sense of well-being of elderly individuals. It might be because elderly individuals create meaningful connections, acquire social recognition, and boost their feelings of well-being via regular discussion with their peers while watching gateball tournaments. Or older individuals may gain from a richer immersion experience due to their ongoing gateball sports intake, prompting needs to be satisfied, emotions to collect, and a steady feeling of optimism for life, which enhances well-being.

### 6.2. Analysis of the Intermediary Role of Social Capital

This research developed a mediation model for the influence of gateball sports engagement on elderly people’s well-being, with social capital playing a 47.84% mediating role. In economics, social capital is defined as a relational commodity that is a driving element in improving well-being. As a result, some researchers describe relationship goods as the emotional expression of interpersonal connections, implying that they have endogenous dynamics. However, this definition is based on reciprocal requirements such as long-term companionship, emotional support, social identity, community membership, and so on. Relational commodities are produced as a consequence of social bonding to address various social requirements, i.e., as a result of social contact. Playing gateball and watching gateball contests are social activities that may help build relationships. As a result, playing gateball is one of the most acceptable methods to make and consume relationship commodities. Involvement in gateball sports allows for the development of social connection, social trust, and social participation. Participation in gateball, particularly in leisure time, provides for social relationships and helps people develop communication and collaboration skills, improving their quality of life. Furthermore, some studies have recognized social contact as an endogenous driver of goalball involvement, and this endogenous incentive is especially prominent among active goalball exercisers and goalball spectators. Active gateball participation can contribute to accumulating social capital by interacting with others through gateball exercises and exploring new social relationships while consolidating existing ones, strengthening a sense of community belonging and promoting social cohesion through gateball volunteering, supporting social identity, and promoting social integration through gateball match attendance. Relational goods emerge from the connection itself, which cannot be created by the person and must thus be shared with others. Individuals, for example, benefit from links when they watch a goalball athletic event and obtain pleasure from discussing their opinions with others. Consumption and production are relative in this context, and involvement in gateball, while “creating” relational benefits, may lead to circumstances of “consumption” for sports players, meeting individual and societal needs and boosting well-being. The significance of social capital in mediating the well-being of elderly people via gateball involvement seems reasonable. This could be because elderly people bond with one another through gateball activities, initially forming a social network based on social trust,’ consolidating social relationships through continuous social interaction, changing social roles and social positioning, and thus improving their sense of well-being. This might also be because, by sharing the experience of watching gateball contests, elderly people create a positive sports environment and broaden their social involvement, improving their sense of belonging and identity in society and boosting their sense of well-being.

### 6.3. Pathway Interpretation of Positive Well-Being

This research divided the paths of how gateball activities may improve the well-being of elderly people into four models based on the seven categories.

Model A’s defining characteristics are “older age” and “sports activity”. As individuals become older, they may become increasingly concerned with how to enhance their physical and mental health and well-being via gateball engagement.

The critical component of Model B is social involvement. Participation in a gateball club may allow elderly people to broaden their social network and earn mutual recognition and trust from club members, leading to a stronger feeling of well-being.

The essential component of Model C is sports involvement. This is most likely because older individuals concentrate on the experience of playing gateball, which leads to a feeling of well-being. On the one hand, engaging in gateball activities provides elderly people with the thrill of triumph and the motivation of failure; on the other hand, watching gateball competitions and consuming gateball sports provides them with psychological gratification.

Model D’s essential components are “low age”, “social trust”, and “social contact”. This is probably because elderly people enjoy the social side of gateball activities, which suit elderly people’s leisure and social demands.

## 7. Study Conclusions and Limitations

### 7.1. Conclusions

SEM demonstrates that:Gateball sport participation can improve elderly people’s well-being.Gateball sport participation can improve elderly people’s social capital (social trust, social participation, and social interaction).Social capital can enhance the well-being of older gateball participants.Social capital can partially mediate the effect of gateball sport participation on elderly people’s well-being.

According to the fsQCA, four patterns in gateball may improve the well-being of elderly people.

### 7.2. Limitations

First, it was challenging to use demographic characteristics effectively because the sample’s age distribution placed nearly half of the population between 60 and 65 years old, and the pre-questionnaire did not include information about elderly people’s health. These factors may affect sports participation, social capital, and overall well-being. Future research should suitably enlarge the population sample size, enhance the application of demographic features, explain the interaction between demographic characteristics and factors, and reinforce the dependability of the study results.

Second, the research region of this work is Hunan Province, China, which has geographical restrictions that make generalizing the study’s conclusions problematic. To increase the generalizability of the research results, the cross-sectional study region should be enlarged in the future to include representative cities such as Beijing, Shanghai, Guangzhou, and Hangzhou.

Third, the age factor impacts the pleasure of older individuals, but the method by which it does so is unclear. It might be because as individuals become more senior, their thinking and experience settle, and they can adjust effectively to the present condition in which their objectives are unfulfilled. In the future, qualitative research on older individuals of various ages may utilize rooting theory to analyze their views of happiness across multiple generations and expand the theoretical elaboration of study results.

Fourth, since this is cross-sectional research, it is impossible to rule out reverse causation between sports participation and well-being, social capital and well-being, and sports participation and social capital. For example, it is more likely to enhance well-being among elderly people in sports social organizations that lack communication and mutual support. More fine-grained intervention studies and longitudinal experiments are therefore required in the future to clarify the causal links between sports engagement, social capital, and well-being, as well as to increase the usefulness of study results.

### 7.3. Recommendations

To begin, focusing on the concept of high-quality gateball leisure, exploring diverse gateball experiences, strengthening the wisdom of gateball venues, promoting the mutual integration of offline gateball exercise and online gateball communication, improving the quality of gateball public service provision, forming a good gateball participation atmosphere, and further encouraging elderly people to participate in gateball activities are recommended.

Second, we should aggressively promote and popularize gateball activities, increase the number of individuals engaging in gateball activities, and emphasize the benefits of gateball activities on the elderly’s quality of life, well-being, and physical and mental health.

Third, as a result of the new crown pneumonia pandemic, we should increase the construction of community gateball courts and strengthen neighborhood spaces to improve elderly people’s sense of identity and belonging to the community, as well as to promote the accumulation and mobilization of social capital.

## Figures and Tables

**Figure 1 ijerph-19-12254-f001:**
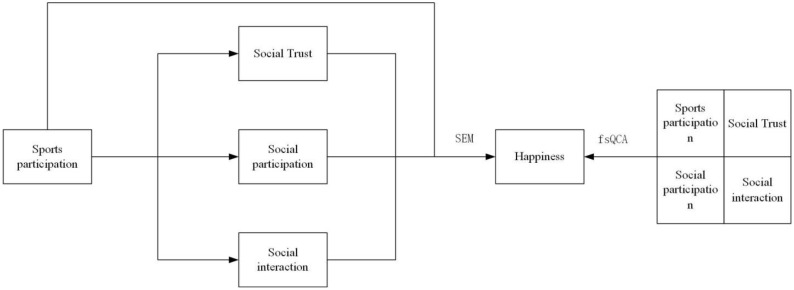
A theoretical model of the influence of sports gateball engagement on elderly people’s well-being.

**Figure 2 ijerph-19-12254-f002:**
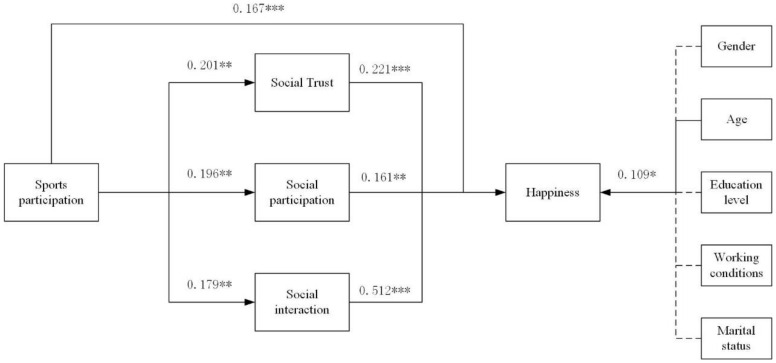
Structural equation modeling of the influence of sports engagement on well-being. Note: *** *p* < 0.001, ** *p* < 0.01, * *p* < 0.05.

**Table 1 ijerph-19-12254-t001:** Demographic features of the study sample.

Indicators	Options	Percentage/%
Gender	Men	46.3
	Women	53.7
Age	“60≤” and “<65”	53.7
	“65≤” and “<70”	39.4
	“≥70”	6.9
Education level	Primary school and below	63.7
	Junior High School	21.9
	High School	10.7
	University and above	3.7
Working conditions	Employment	35.9
	Retirement	63.8
	Temporarily unemployed	0.3
Marital status	Married	84.6
	Divorce	8.1
	Single	5.7
	Bereaved spouse	1.6
Weekly gateball participation frequency (Gateball exercise)	“≤2”	31.6
	“3≤” and “≤4”	44.9
	“5≤” and “≤6”	17.1
	“=7”	6.4

**Table 2 ijerph-19-12254-t002:** Table of confidence validity tests.

Latent Variables	Observation Variables	Standard Load	Cronbachα	CR	AVE
Sports participation	How hard do you practice gateball?	0.845 ***	0.874	0.9431	0.7353
	What time of day will you be playing gateball?	0.921 ***			
	Do you often take part in gateball events?	0.839 ***			
	Do you often watch gateball events?	0.789 ***			
	How much do you spend annually on gateball activities?	0.796 ***			
	How well do you keep up with news on gateball events?	0.943 ***			
Social Trust	Do you believe the majority of people are trustworthy?	0.736 ***	0.728	0.8016	0.5742
	Do you feel at home in your community?	0.743 ***			
	Do you feel secure crossing the street at night?	0.793 ***			
Social participation	Do you belong to any clubs or local organisations?	0.775 ***	0.743	0.7983	0.5694
	Have you participated in any neighborhood or local community events lately?	0.785 ***			
	Would you be interested in helping a local nonprofit organisation?	0.701 ***			
Social interaction	When you shop locally, do you run across friends or familiar faces?	0.703 ***	0.713	0.7807	0.5432
	Do your buddies provide a hand when you’re in need?	0.778 ***			
	Whom did you chat with the most yesterday?	0.728 ***			
Happiness	Is your life joyful, satisfying, and enjoyable? (in the past month)	0.842 ***	0.795	0.861	0.6738
	How much do you value your health? (in the past month)	0.821 ***			
	How do you feel about your spirit, vitality, and vigour? (in the past month)	0.799 ***			

Note: *** *p* < 0.001.

**Table 3 ijerph-19-12254-t003:** Model fitness test table.

Adaptation Indicators	Recommended Value	Fitted Values
χ^2^	The smaller the better	171.322
χ^2^/DF	<3.0	1.839
RMSEA	<0.08	0.027
NFI	>0.9	0.951
IFI	>0.9	0.988
CFI	>0.9	0.971
GFI	>0.9	0.921
AGFI	>0.8	0.929
TLI	>0.9	0.971
PNFI	>0.5	0.741
PCFI	>0.5	0.783

**Table 4 ijerph-19-12254-t004:** Correlation coefficients of variables.

Variables	Sports Participation	Social Trust	Social Participation	Social Interaction	Happiness	M (SD)
Sports participation	1	0.492 ***	0.311 ***	0.254 ***	0.198 ***	5.27 (0.74)
Social Trust		1	0.278 ***	0.221 **	0.208 ***	4.27 (1.07)
Social participation			1	0.237 ***	0.307 ***	3.89 (0.71)
Social interaction				1	0.201 **	3.72 (0.69)
Happiness					1	9.91 (1.27)

Note: ** *p* < 0.01, *** *p* < 0.001.

**Table 5 ijerph-19-12254-t005:** Shows the outcomes of each path’s tests.

Path Relationships	Standardized Factor Loadings	CR Value	*p*	Test Results
Sports participation→social trust	0.201	3.027	**	Support
Sports participation→Social participation	0.196	3.168	**	Support
Sports participation→Social interaction	0.179	3.011	**	Support
Sports participation→happiness	0.167	5.294	***	Support
Social trust→happiness	0.221	4.183	***	Support
Social participation→happiness	0.161	3.166	**	Support
Social interaction→happiness	0.512	11.539	***	Support

Note: ** *p* < 0.01, *** *p* < 0.001.

**Table 6 ijerph-19-12254-t006:** Path test results for the mediating function of Bootstrap.

Paths	Effect Value	Boot SE	*p*	Bias-Corrected 95% CI		Percentile 95% CI	
				Lower Limit	Upper Limit	Lower Limit	Upper Limit
Overall effect	0.324	0.045	***	0.249	0.422	0.251	0.423
Direct effects	0.169	0.034	***	0.101	0.221	0.099	0.235
Indirect effects	0.155	0.027	***	0.094	0.224	0.098	0.225
Sports participation→Social trust→Happiness	0.029	0.012	**	0.008	0.061	0.006	0.059
Sports participation→Social participation→Happiness	0.027	0.009	**	0.008	0.053	0.005	0.049
Sports participation→Social interaction→Happiness	0.028	0.011	**	0.013	0.059	0.011	0.053

Note: ** *p* < 0.01, *** *p* < 0.001; 5000 Bootstrap Sample.

**Table 7 ijerph-19-12254-t007:** Examines the circumstances required for happiness.

Antecedent Variables	Consistency	Coverage
Age	0.744	0.781
~Age	0.522	0.671
Sports participation	0.823	0.819
~Sports participation	0.461	0.627
Social trust	0.816	0.825
~Social trust	0.478	0.639
Social participation	0.814	0.818
~Social participation	0.489	0.648
Social interaction	0.834	0.832
~Social interaction	0.473	0.638

**Table 8 ijerph-19-12254-t008:** Results of the analysis of conditional groups of positive well-being in older persons.

Conditions	Configuration						
	A1	A2	B1	B2	C1	D1	D2
Age	●	●				○	○
Sports participation	●	●	•		●	•	•
Social trust		•	•	•	•	●	●
Social participation	•	•	●	●		•	
Social interaction	•			•	•	●	●
Consistency	0.939	0.911	0.863	0.917	0.953	0.961	0.883
Original coverage	0.337	0.392	0.253	0.375	0.509	0.412	0.454
Unique coverage	0.019	0.042	0.027	0.045	0.041	0.051	0.062
Total consistency	0.917						
Total coverage	0.893						

Note: “●” means the core condition is present, “○” means the condition is not present, “•” means the marginal condition is present, “blank” means the condition is at a medium level.

## Data Availability

Not applicable.

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
