# Peer review of "The Effect of Playing Gateball Sports on Older Chinese People’s Wellbeing in the Context of Active Aging—Based Mediation of Social Capital"

_ijerph, 2022, doi:10.3390/ijerph191912254_

Round 1

Reviewer 1 Report

This article analyse a topical issue given that the global population is getting older, therefore requiring a healthy aging people. To promote health sports are vital. The authors give several statistics regarding the ageing of Chinese population and then they focus on the link between sport engagement of older people and social capital.

 The structure of the article is chosen correctly and it exhibits a logical concern. The methodology is carefully and deeply applied. However, I have a few recommendations about the article:

The hypotheses shall be formulated in an affirmative manner. Therefore, I recommend to remove the word “may” in line 118. The same recommendation for hypothesis 3 stated in line 255. When referring to COVID 19 authors should be consistent and identify it as pandemic and not epidemic, as in line 284. Regarding table 2, I was not able to find the variables’ definitions.

In my opinion the impact of COVID 19 in the ability and willing of elder people to be involved in sports activity would be important., In addition, I suggest to add some latest references from 2021, 2022, which analyses the impact of the COVID 19 pandemic.

Author Response

List of responses

Dear Editors and Reviewers

Thank you for your letter and for the reviewers' comments concerning our manuscript entitled "The effect of playing gateball sports on older Chinese people's wellbeing in the context of active aging — Based mediation of social capital" (ID:ijerph-1908332). Those comments are all valuable and very helpful for revising and improving our paper, as well as the essential guiding significance to our research. We have studied the comments carefully and made a correction that we hope meets with approval. Revised portions are marked in red on the paper. The significant corrections in the paper and the responses to the reviewer's comments are as follows:

Responds to the reviewer's comments:

Reviewer #1:

  1. Response to comment: The hypotheses shall be formulated in an affirmative manner. Therefore, I recommend to remove the word “may” in line 118. The same recommendation for hypothesis 3 stated in line 255. When referring to COVID 19 authors should be consistent and identify it as pandemic and not epidemic, as in line 284. Regarding table 2, I was not able to find the variables’ definitions.

Response: Thank you very much for your insightful remarks. In response to your concerns, I have made the following changes: (1) I have deleted the term "potential" from Hypotheses 2 and 3, and (2) I have changed the word "epidemic" throughout the text to "pandemic," and (3) I have provided clarification to the definitions in Table 2.

  1. Response to comment: In my opinion the impact of COVID 19 in the ability and willing of elder people to be involved in sports activity would be important., In addition, I suggest to add some latest references from 2021, 2022, which analyses the impact of the COVID 19 pandemic.

Response: Thank you for your feedback. It is crucial to note that the influence of COVID-19 on older people should be considered while reviewing the research since we first neglected to do so. We have taken note of your insightful remarks and have included pertinent material in the introduction and literature review sections to clarify the connection between older people's well-being during COVID-19, sports engagement, and social capital. Additionally, the references after the text have been rearranged to make them more straightforward.

Special thanks to you for your good comments.

Other changes:

  1. To make the synopsis simpler for readers to grasp, we reorganized it.

We did our most complicated to enhance the text and made some adjustments. These modifications will not affect the paper's content or structure. Furthermore, instead of listing the modifications, we highlighted them in red in the amended document. We sincerely appreciate the editors/reviewers' dedicated effort and hope any modifications will be addressed. Thank you for your comments and recommendations once again.

Yours sincerely

Xinze Li

Reviewer 2 Report

While the topic is of value to the aging population, there are some over assumptions made concerning what can be taken from this study and there are multiple aspects of the study that are unclear and seem to lack focus. To begin with the title needs to be more specified as to what is clearly studied. While "sports" is indicated within the title, this is too generic when the population targeted was only those participating at gateball events. This is an overly generic term when additional sports were not targeted for participation. The term "engagement" within the title is also vague as Table 1 indicates "participation". Is that one in the same when it comes to how the authors define these terms, and if so, what does that mean? Could observing mean "engagement" and/or "participation" or does the individual need to be actively doing the sport? Finally, the population was 60 years or older, but the title indicates "older adults", which could include a population under the age of 60 years. The population targeted would refer to a "geriatric" population or other terms such as "senior" or "elderly". Truly, in the end, the population sampled does not seem to represent all older Chinese adults that are involved in all sports, and thus, the title needs to be more specific staying focused on what population participated.

As for the literature review, section 2, it needs to be reduced as, at times, it seems repetitive. Quotes are utilized too often within this section. The authors need to utilize their own words and interpret the literature appropriately. Sections 2.2.1 and 2.2.2 need to be combined with the two tied together directly relating aging to social capital. Switch sections 2.3 and 2.4 in the order these two topics are given, although leave lines 257-272 at the end of these two sections.

As for the methods, was this study reviewed by an institutional ethics review board for human studies? This research must be reviewed by this type of review board and this information must be given within this methods section. How was confidentiality ensured for participants of this study and was that clearly stated for participants within the questionnaire? 

What is gateball and why was this sport selected for this study? Why is this an important sport for the geriatric Chinese population? This information needs to be given in the introduction. Since gateball was the targeted sport, this needs to not only be given in the title, but also the introduction needs to be targeted on this specific sport and the value of this activity. If the authors want to give a more generic discussion focused on all sports, then, the sampled population needs to be much larger and recruitment needed to be done at various other activities besides just gateball. Furthermore, since recruitment was done only at one sporting activity, gateball, Table 1 needs to be specific as to the frequency of the participation of that activity. Also, "participation" should have been clearly defined to the questionnaire participants, and if it was, what was that definition? How involved is "participation" such as the length of the activity per day and the intensity. Sports participation should be divided up between gateball and other sports and those other sports need to be identified as to whether they are truly sports or recreational activities like taking the dog for a walk compared to running a marathon or playing a baseball game. Each sport can have a different physical impact on an aging adult, particularly depending on the intensity level. More so, for individuals that may be participating in the sport of horseback riding, there have been numerous studies done discussing the therapeutic impact, physical, emotional, and cognitive, and thus, these individuals may have more of a positive response when it comes to "well-being" than other participants. Therefore, a clear, more targeted approach concerning the sport needs to be taken for this study. 

Beyond the sport-related activities of the participants, there is a concern as to what was their health and mental status at the time of this study. This study was carried out during the COVID pandemic, and thus, enduring this health condition could impact not only activities, but well-being. In addition, individuals diagnosed with mental health conditions would also have a different response to those that do not. A clear understanding of their current health status, both physically and mentally, beyond their own perceived, self-reported well-being would be useful where there is clear indication of any diagnosed conditions they are coping with at the time of the questionnaire.

How was the questionnaire distributed? Was it paper only format? Did researchers alone responsible for recruitment and was that done solely at the gateball events? Was anything offered to participants in return of participation such as money? Was participation voluntary? What was the response rate, meaning how many from each event participated in the survey? If paper format, how many were passed out versus those returned? What was the inclusion and exclusion criteria?

As for the discussion section, the authors use interchangeably "sports involvement", "physical activity", "sports engagement", "sports participation", "indirect" or "direct" "sports engagement", but it's confusing as to what was actually studied from the demographics given for the questionnaire participants as to whether these were truly active elderly adults or just one's that observed sports. Most of the focus seems to be on active geriatric adults, but a significant portion of the discussion focuses on just those that watch a sport. Again, the definition of "engagement" or "participation" is unclear and leaves it hard to determine what conclusions can be drawn from the population sampled. If looking at both "direct" and "indirect" involvement, that really needed to be set up as two sample populations followed by statistical analysis looking at the differences between the two. Putting the two groups together, and then, trying to make conclusions on both aspects of "engagement" can not be supported at this time off of the data presented. 

Section 6.3 makes some broad assumptions and conclusions that does not appear to be supported by the data presented nor previous research given. The generalizations made are not supported, especially when the focus appears to be specifically targeted on gateball, but the majority of the discussion has been on generalized sports. Furthermore, within this section the authors begin to utilize the term "seniors", but throughout most of the manuscript including the title, the focus has been "older adults". Terminology needs to be consistent throughout, and yes, the use of the term "senior", "geriatric", and/or "elderly" would be more appropriate than "older adults". 

Finally, for section 7, conclusions need to be more targeted towards what was actually measured. Broad conclusions can not be made off of the sample population utilized. Yes, the authors are correct in the limitations given, particularly as it pertains to the sample population, and thus, discuss how the researchers may have addressed these limitations or recommendations on what limited conclusions can be drawn due to these limitations. In reading these limitations without given any justification on why this research should stand alone despite these limitations, it is hard to make any conclusions from this study. Lastly, the recommendations should be targeted on what can be taken away from the study and bringing in the discussion of "legal and regulatory systems" needing to be "improved" when this had not been discussed previously and was not a focus of the study seems off-base from the study objectives.

Author Response

Please see the attachment:

Round 2

Reviewer 2 Report

The authors should be first commended on their extensive revisions that were completed within a short timeframe. While this manuscript would be better served if all of the revisions could have been more thoroughly addressed, but without completely redoing the study itself, the authors did an adequate job addressing the reviewers concerns within these limitations. Nevertheless, a few additional revisions should be addressed before publication.

Within the introduction, the phrase of "some studies" or "some research" is utilized quite frequently, and while the repetitive nature alone should be discouraged, the vagueness of this statement should be avoided, especially when multiple sentences utilizing these phrases only cited one publication. The term "some" should equate to multiple studies, and thus, references within such sentences should include multiple citations. In addition, it would be more impactful to the reader to have specific discussion of these studies than just categorizing them together as "some".

The authors are commended for the addition of information about gateball within the introduction and within the title itself, but the authors did not include within the objective statement, hypothesis, nor within the abstract this more focused direction of gateball, and thus, the discussion of gateball needs to be included within these areas. Also, while gateball is included within section 7, specifically within the conclusions, additional focus needs to be added specific to this sport within the previous paragraphs of the discussion. 

While the authors tried to address the reviewer's concerns as it related to terminology associated with the subjective use of the term "older adults" within the previous manuscript, switching "adults" to "people" does not objectively identify the issue of what is objectively considered "older". In fact, "people", instead of "adults" leaves the interpretation even broader. The authors are advised to utilize either "geriatric" or "elderly", even "aging adults" and "older adults" are acceptable if age range is clearly specified giving the reader context to the meaning. Within most Internet searches the terms "geriatric" or "elderly" reflect 65 years or older, and thus, age within the manuscript does not need to be specified when using such terms. However, if the authors want to be more specific, they can either include the age such as saying "older adults over the age of 60" or they can use terms such as "late elderly" to refer to individuals older than 75 or "early elderly" to refer to individuals in their early 60's.

Finally, while section 7.3 within the previous manuscript seemed off message from the results presented within that version of the manuscript, the authors are advised that instead of completely removing this section, they should look at revising it. The discussion of limitations within section 7.2 gives some suggestion of what might be recommended for future research in this area, making for a good discussion of recommendations within a revised section 7.3. Authors can can even look at offering up suggestions for government programs for the geriatric that follow what was learned within this study. Specifically, would additional sports need to be researched besides gateball or should the government look at establishing gateball programs? What about long term studies? This section on recommendations does not need to be an extensive section of the discussion, but would be useful after the section 7.2 on limitations so that the authors can refocus the reader on what can be taken from this study. Seems to discredit the work within this manuscript to end with limitations. Again, however, similar to what was mentioned in the previous review, the authors are advised to stay focused on the study and specifically what was directly learned from the results.  

Author Response

List of responses

Dear Editors and Reviewers

Thank you for your letter and for the reviewers' comments concerning our manuscript entitled "The effect of playing gateball sports on older Chinese people's wellbeing in the context of active aging — Based mediation of social capital" (ID:ijerph-1908332). Those comments are all valuable and very helpful for revising and improving our paper, as well as the essential guiding significance to our research. We have studied the comments carefully and made a correction that we hope meets with approval. Revised portions are marked in red on the paper. The significant corrections in the paper and the responses to the reviewer's comments are as follows:

Responds to the reviewer's comments:

Reviewer #1:

  1. Response to comment: Within the introduction, the phrase of "some studies" or "some research" is utilized quite frequently, and while the repetitive nature alone should be discouraged, the vagueness of this statement should be avoided, especially when multiple sentences utilizing these phrases only cited one publication. The term "some" should equate to multiple studies, and thus, references within such sentences should include multiple citations. In addition, it would be more impactful to the reader to have specific discussion of these studies than just categorizing them together as "some".

Response: Thank you so much for your insightful remarks. We changed it per your ideas, re-condensing the phrases and including several references to the literature.

  1. 2. Response to comment:The authors are commended for the addition of information about gateball within the introduction and within the title itself, but the authors did not include within the objective statement, hypothesis, nor within the abstract this more focused direction of gateball, and thus, the discussion of gateball needs to be included within these areas. Also, while gateball is included within section 7, specifically within the conclusions, additional focus needs to be added specific to this sport within the previous paragraphs of the discussion. 

Response: Thank you so much for your insightful remarks. First, as you indicated, we added gateball to sections 2.1 and 2.4; second, we inserted a standard application comment for "sports participation" in section 3, lines 268-269, to minimize reader misunderstanding; and third, we rewrote section 6 to concentrate on the gateball program.

  1. 3. Response to comment:While the authors tried to address the reviewer's concerns as it related to terminology associated with the subjective use of the term "older adults" within the previous manuscript, switching "adults" to "people" does not objectively identify the issue of what is objectively considered "older". In fact, "people", instead of "adults" leaves the interpretation even broader. The authors are advised to utilize either "geriatric" or "elderly", even "aging adults" and "older adults" are acceptable if age range is clearly specified giving the reader context to the meaning. Within most Internet searches the terms "geriatric" or "elderly" reflect 65 years or older, and thus, age within the manuscript does not need to be specified when using such terms. However, if the authors want to be more specific, they can either include the age such as saying "older adults over the age of 60" or they can use terms such as "late elderly" to refer to individuals older than 75 or "early elderly" to refer to individuals in their early 60's.

Response: Thank you very much for your helpful advice. In response to your feedback, we have changed the phrasing used in the post for older people to "elderly people" instead of "older people."

  1. 4. Response to comment:Finally, while section 7.3 within the previous manuscript seemed off message from the results presented within that version of the manuscript, the authors are advised that instead of completely removing this section, they should look at revising it. The discussion of limitations within section 7.2 gives some suggestion of what might be recommended for future research in this area, making for a good discussion of recommendations within a revised section 7.3. Authors can can even look at offering up suggestions for government programs for the geriatric that follow what was learned within this study. Specifically, would additional sports need to be researched besides gateball or should the government look at establishing gateball programs? What about long term studies? This section on recommendations does not need to be an extensive section of the discussion, but would be useful after the section 7.2 on limitations so that the authors can refocus the reader on what can be taken from this study. Seems to discredit the work within this manuscript to end with limitations. Again, however, similar to what was mentioned in the previous review, the authors are advised to stay focused on the study and specifically what was directly learned from the results.  

Response:Thank you very much for your helpful advice. We took your suggestions and modified section 7.3 of the article to include pertinent remarks from the study's results in the hopes of increasing the well-being of elderly people via gateball activities.

We did our most complicated to enhance the text and made some adjustments. These modifications will not affect the paper's content or structure. Furthermore, instead of listing the modifications, we highlighted them in red in the amended document. We sincerely appreciate the editors/reviewers' dedicated effort and hope any modifications will be addressed. Thank you for your comments and recommendations once again.

Yours sincerely

Xinze Li